# Biomarkers in a socially exchanged fluid reflect colony maturity, behavior, and distributed metabolism

**Sanja M Hakala[1], Marie-Pierre Meurville[1], Michael Stumpe[2], Adria C LeBoeuf[1]\***

[1]Department of Biology, University of Fribourg, Fribourg, Switzerland; [2]Metabolomics and Proteomics Platform, Department of Biology, University of Fribourg, Fribourg, Switzerland

**Abstract** In cooperative systems exhibiting division of labor, such as microbial communities, multicellular organisms, and social insect colonies, individual units share costs and benefits through both task specialization and exchanged materials. Socially exchanged fluids, like seminal fluid and milk, allow individuals to molecularly influence conspecifics. Many social insects have a social circulatory system, where food and endogenously produced molecules are transferred mouth-to-mouth (stomodeal trophallaxis), connecting all the individuals in the society. To understand how these endogenous molecules relate to colony life, we used quantitative proteomics to investigate the trophallactic fluid within colonies of the carpenter ant *Camponotus floridanus*. We show that different stages of the colony life cycle circulate different types of proteins: young colonies prioritize direct carbohydrate processing; mature colonies prioritize accumulation and transmission of stored resources. Further, colonies circulate proteins implicated in oxidative stress, ageing, and social insect caste determination, potentially acting as superorganismal hormones. Brood-caring individuals that are also closer to the queen in the social network (nurses) showed higher abundance of oxidative stress-related proteins. Thus, trophallaxis behavior could provide a mechanism for distributed metabolism in social insect societies. The ability to thoroughly analyze the materials exchanged between cooperative units makes social insect colonies useful models to understand the evolution and consequences of metabolic division of labor at other scales.

**\*For correspondence:**
adria.leboeuf@unifr.ch

**Competing interest:** The authors declare that no competing interests exist.

## Introduction

In the course of social evolution, related organisms have formed cooperative entities such as multicellular organisms or groups of social animals (*Szathmáry, 2015*; *Queller, 1992*; *Hamilton, 1963*). In social animal groups, collective decisions on movement, reproduction and even development are needed for survival (*Miller et al., 2013*; *Couzin, 2009*). Some social groups have taken this coordination to a very high level: social insect societies develop and function as a single unit instead of as competing individuals, as 'superorganisms' paralleling the development of multicellular organisms as a single unit rather than as a set of competing cells (*Johnson and Linksvayer, 2010*; *Boomsma and Gawne, 2018*).

In these superorganismal societies, reproductive queens and males function as the germline, and workers as the soma. Similarly to different tissues in multicellular organisms, workers can be further specialized and exhibit division of labor across different behavioral and morphological castes (*Bourke, 2011*). While morphological castes are determined during development, the behavioral caste of an individual worker typically changes during its lifetime. At the beginning of their adult life, workers specialize inside the nest as nurses focusing on brood care, and as they age, they switch to foraging outside of the nest (*Huang and Robinson, 1996*). Social insect

**eLife digest** Division of labor is essential for cooperation, because groups can achieve more when individuals specialize in different tasks. This happens across the natural world, from different cells in organisms performing specific roles, to the individuals in an ant colony carrying out diverse duties. In both of these systems, individuals work together to ensure the survival of the collective unit – the body or the colony – instead of competing against each other. One of the main ways division of labor is evident within these two systems is regarding reproduction. Both in the body and in an ant colony, only one or a few individual units can reproduce, while the rest provide support. In the case of ant colonies, only queens and males reproduce, while the young workers nurse the brood and older workers forage for food.

This intense cooperation requires close communication between individual units – in the case of some species of ants, by sharing fluids mouth-to-mouth. These fluids contain food but also many molecules produced by the ants themselves, including proteins. Given that both individuals and the colony as a whole change as they age – with workers acquiring new roles, and new queens and males only reared once the colony is mature – it is likely that the proteins transmitted in the fluid also change.

To better understand whether the lifecycles of individuals and the age of the colony affect the fluids shared by carpenter ants *Camponotus floridanus*, Hakala et al. examined the ant-produced proteins in these fluids. This revealed differences in the proteins shared by young and mature colonies, and young nurse ants and older forager ants. In young colonies, the fluids contained proteins involved in fast sugar processing; while in mature colonies, the fluids contained more proteins to store nutrients, which help insect larvae grow into larger individuals, like queens. Young worker ants, who spend their time nursing the brood, produced more anti-aging proteins. This may be because these ants are in close contact with the queen, who lives much longer than the rest of the ants in the colony. Taken together, these observations suggest that ants divide the labor of metabolism, as well as work and reproduction.

Dividing the labor of metabolism among individuals is one more similarity between ants and the cells of a multicellular organism, like a fly or a human. Division of labor allows the sharing of burden, with some individuals lightening the load of others. Understanding how ants achieve this by sharing fluids could shed new light on this complex exchange at other scales or in other organisms. By matching proteins to life stages, researchers have a starting point to examine individual molecules in more detail.

---

colonies also go through life stages. Young colonies have an initial growth phase where they solely produce one type of worker, and only later in their life cycle they may produce more specialized worker castes and finally, males and queens (*Wilson, 1971*). The switch to reproductive phase is a major life-history transition at the colony level, and connected to female caste determination. In social Hymenoptera, determination of whether a female larva develops into a queen or a worker, and what kind of worker exactly, is controlled by intricate differences of gene expression of the same female genome, guided primarily by environmental factors, in particular nutrition and social cues, sometimes partially influenced by genetics (*Wheeler, 1986*; *Anderson et al., 2008*; *Rajakumar et al., 2018*; *Schwander et al., 2010*).

Coordinated function of tightly integrated groups such as social insect colonies, and subgroups such as their different castes, has been described as social physiology (*Friedman et al., 2020*), consisting of various behavioral, morphological, and molecular mechanisms that ensure cooperation and inclusive fitness benefits for all group members. As a part of their social physiology, some social insect societies have developed a form of social circulatory system (*Wheeler, 1928*), where nutrition and endogenously produced functional molecules, such as hormones, are transferred mouth-to-mouth from the foregut of one individual to another (*LeBoeuf et al., 2016*; *LeBoeuf et al., 2018*). This social fluid transfer is called stomodeal trophallaxis (*Meurville and LeBoeuf, 2021*). It ensures not only that food is distributed to all adults and larvae within the colony, but also that all individuals of the colony are interconnected through shared bodily fluids. Trophallactic fluid of ants and bees typically contains endogenous proteins involved in digestion, immune defense and developmental regulation (*LeBoeuf et al., 2016*), indicating that this fluid transmits more than food.

Molecular signals are important in controlling the colony life histories and guiding caste determination both at the colony level and at the individual level. Queen pheromones are central signaling molecules acting across individuals (*Kocher and Grozinger, 2011*; *Nijhout and Wheeler, 1982*; *Pamminger et al., 2016*). Juvenile hormone and vitellogenin are central signaling molecules in classical insect development that may also play across-individual roles in some social insects (*LeBoeuf et al., 2016*; *LeBoeuf et al., 2018*; *Scharf et al., 2007*; *Harwood et al., 2019*). Together with fundamental nutrient-response signaling pathways (insulin, TOR), these molecules establish the developmental trajectories of individuals (*Chandra et al., 2018*; *Libbrecht et al., 2013*). In solitary organisms, such molecules are produced and function solely within the organism's own body. In contrast, in social Hymenoptera even the molecules traditionally functioning within-individuals can be secreted to the crop and distributed among the society members through trophallaxis and the social circulatory system (*LeBoeuf et al., 2016*).

Molecular components transmitted through trophallaxis, namely juvenile hormone and juvenile hormone esterase-like proteins, have been shown to influence the development of ant larvae (*LeBoeuf et al., 2016*; *LeBoeuf et al., 2018*). Thus, it is possible that molecules in trophallactic fluid may influence caste determination, similarly to honeybee workers feeding larvae with royal jelly to direct their development toward a queen fate (*LeBoeuf et al., 2016*; *Buttstedt et al., 2014*; *Buttstedt et al., 2016*; *Kamakura, 2011*; *Kucharski et al., 2015*). The molecular functions of trophallactic fluid are still largely unstudied, but it is known, for example, that social isolation changes its composition (*LeBoeuf et al., 2016*), with some protein components of this fluid shifting with social environment. In medicine, such correlations are typically used to define biomarkers for specific conditions and treatments, and often both accurately predict function and provide mechanistic insights (*Strimbu and Tavel, 2010*). We propose that trophallactic fluid could both reflect and affect the social environments of the colony, thus providing important cues for collective decision making. However, it is not yet feasible to study the causes and consequences of the molecular composition of trophallactic fluid, as it is still largely unknown how much and what kind of qualitative and quantitative variation is present.

If indeed trophallactic fluid acts as a form of social circulatory system, managing distributed metabolic processes related to colony maturation, endogenously produced factors should correlate with colony life stages. To test this, we analyzed the trophallactic fluid proteome of the carpenter ant *Camponotus floridanus* at different scales. Our aim is to demonstrate that trophallactic fluid proteomes are filled with biomarkers reflecting biotic and abiotic conditions at both the colony and individual scale.

## Results

We sought to determine whether the endogenously produced proteins present in trophallactic fluid create a robust biomarker-like signature of colony status. To assess this, we analyzed the trophallactic fluid proteomes of colonies at different stages in the colony life-cycle (*Young vs. Mature*), of colonies in natural conditions or kept in the lab (*Field vs. Lab*), and between colonies found on different nearby islands (*East vs. West*) (*Figure 1*; *Supplementary file 1*). Because trophallactic fluid proteins may be differentially expressed, transmitted, and/or sequestered across the social network of a colony, we also analyzed trophallactic fluid proteomes of single individuals in different colony 'tissues' – in-nest workers taking care of brood and out-of-nest workers (*Nurse vs. Forager*).

### Overall proteome variation

Over the 73 colony and 40 single-individual trophallactic fluid samples analyzed, a total of 519 proteins were identified (*Figure 2*). Trophallactic fluid samples contained a set of 27 'core' trophallactic fluid proteins that were present in all samples regardless of life-cycle, life-stage or environmental conditions. Fifty-seven percent of the 519 proteins, we observed were present in less than half of the samples. Even though the most common proteins displayed higher average abundance, across the entire dataset, protein abundance did not correlate with the proportion of samples containing the protein – even proteins present in only a small proportion of the samples in some cases exhibited high abundance (*Figure 2—figure supplement 1*). The overall protein abundance was higher in colony samples relative to single individual samples, reflective of the larger trophallactic fluid volume collected. The number of proteins identified for a given sample correlated with trophallactic fluid

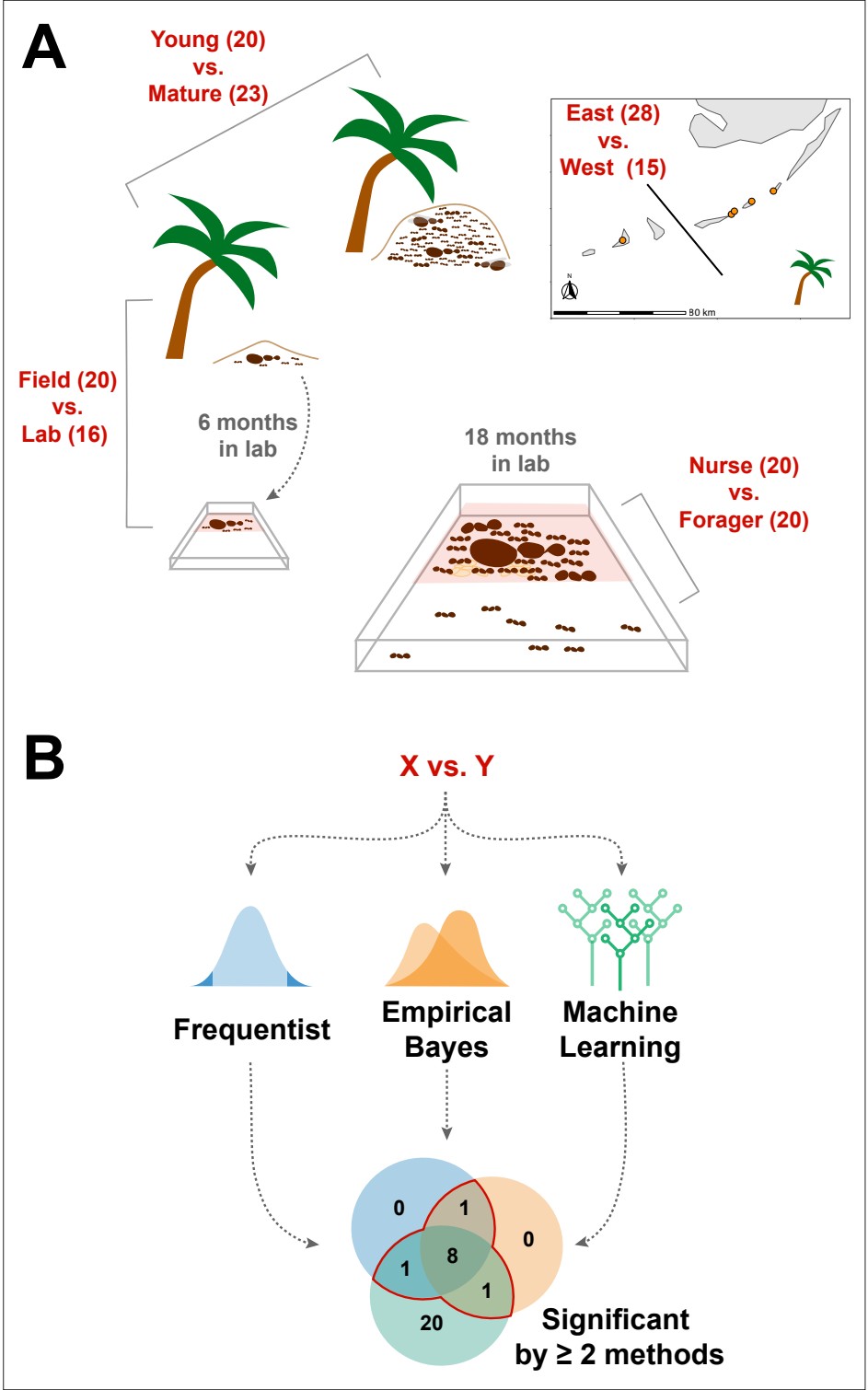

**Figure 1.** Schematic of study design. (**A**) Four comparisons, *Young vs. Mature*, *Nurse vs. Forager*, *Field vs. Lab*, and *East vs. West*, analyzed in this study with sample numbers indicated in parentheses. In all comparisons sample numbers indicate colonies with the exception of *Nurse vs. Forager*, where samples are from single individuals, ten each from four colonies. Palm trees indicate field samples and boxes indicate laboratory samples. (**B**) Schematic of analysis approach to find robustly differing proteins in each comparison. Sample information can be found in *Supplementary file 1*.

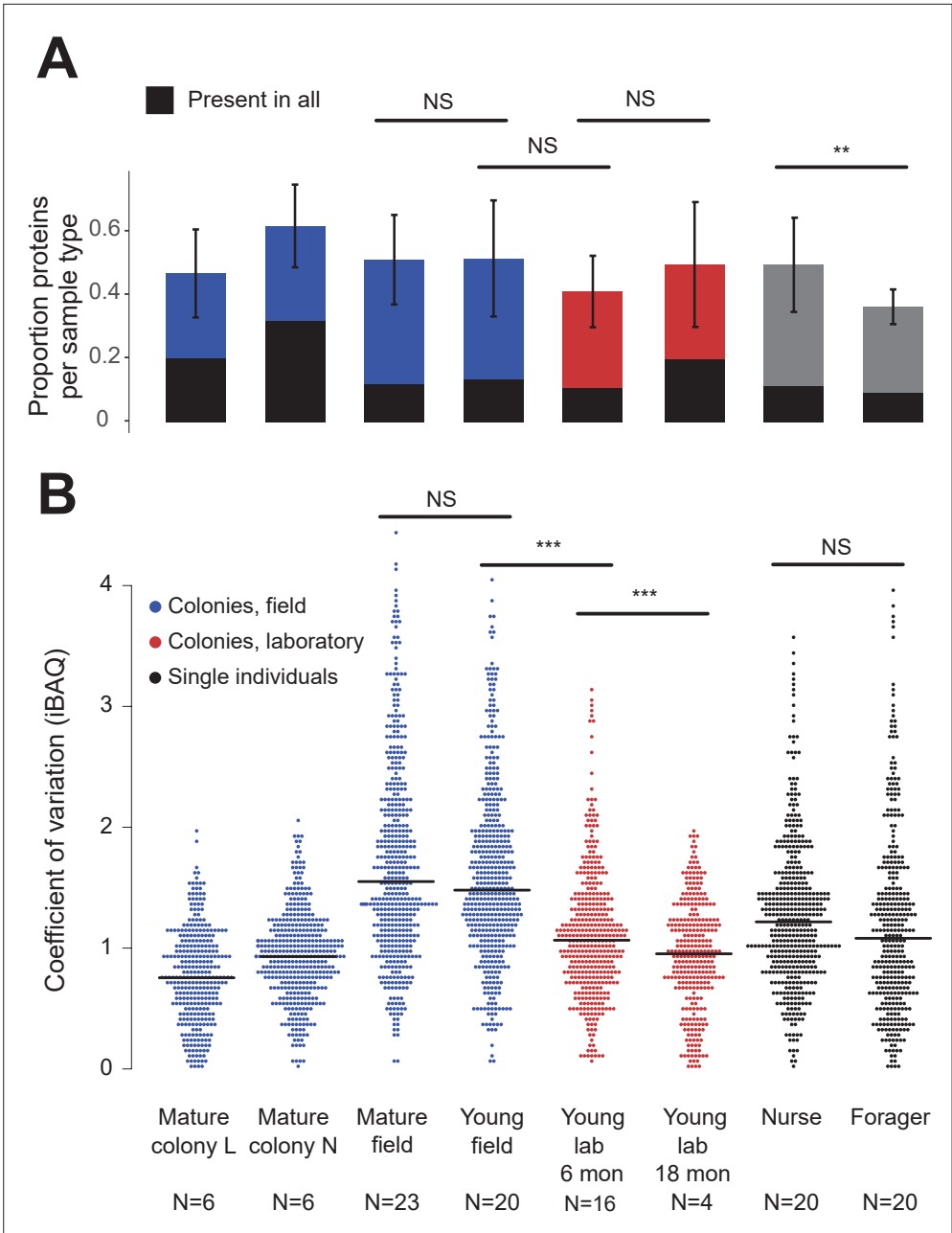

**Figure 2.** Protein presence in trophallactic fluid varies with biotic and abiotic factors. (**A**) Mean ± SD of the proportion of proteins present in samples of a given type. Proportion of proteins present in all samples of a given type are highlighted in black. (**B**) Coefficient of variation (standard deviation/mean), calculated for the iBAQ values greater than zero of all the proteins identified by sample type. Sample sizes per type are given under their names. Mature L and Mature N are mature colonies that were sampled six times to assess within-colony variation in colony samples. Significance of comparisons based on gamma GLM (**A**) or negative binomial GLM (**B**): NS indicated when $p > 0.05$ significant, ** $p < 0.01$, *** $p < 0.001$ (full results in *Figure 2—source data 1*; *Figure 2—source data 2*).

The online version of this article includes the following figure supplement(s) for figure 2:

**Source data 1.** Coefficient of variation by sample type Post-hoc comparisons of gamma GLM on coefficient of variation by sample type.

**Source data 2.** Protein number by sample type Post-hoc comparisons of negative binomial GLM on protein number explained by sample type.

**Figure supplement 1.** Protein abundance and commonness.

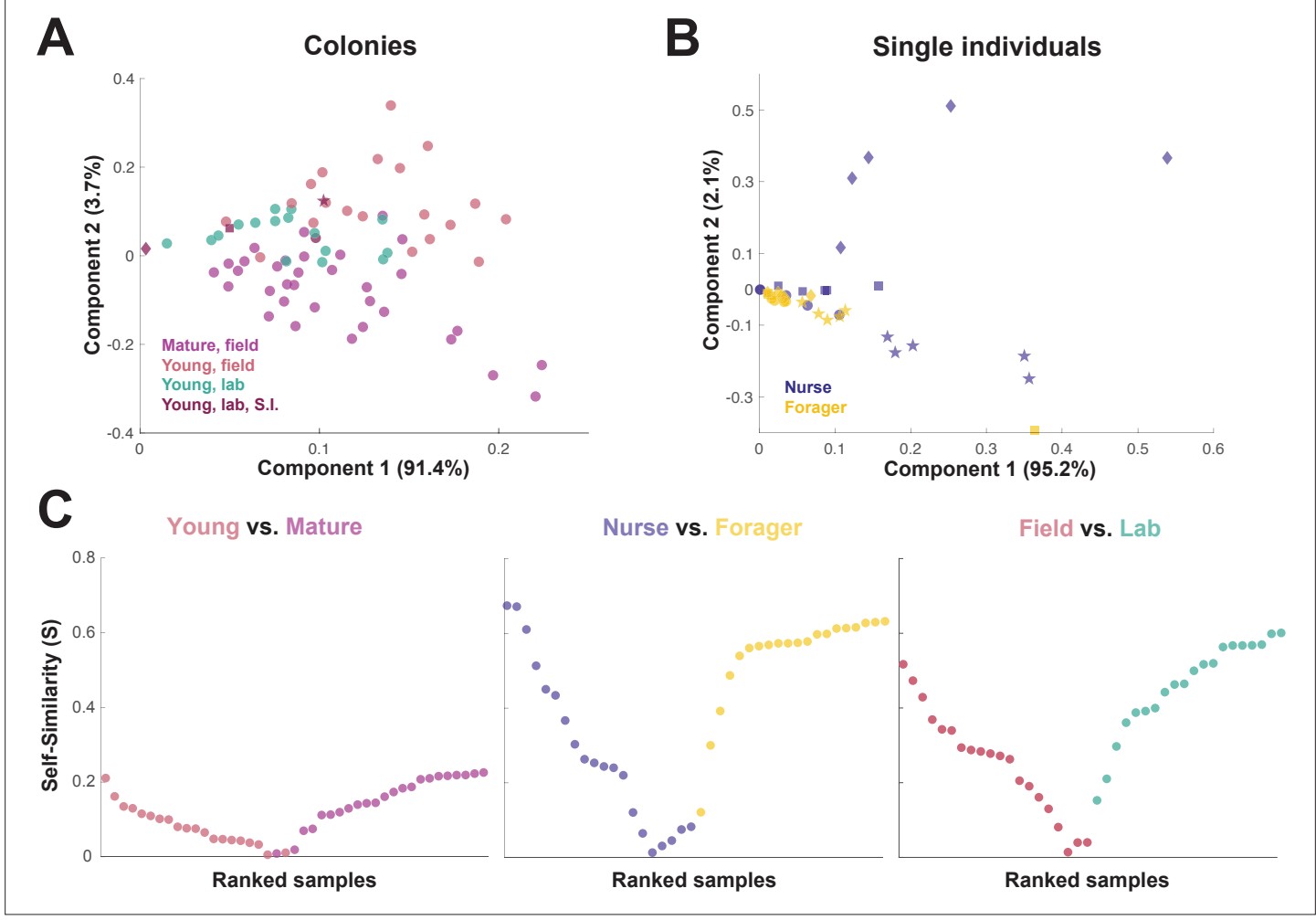

**Figure 3.** Similarity across trophallactic fluid proteome samples of colonies and single individuals. Principal component analysis for all proteins for (**A**) colony samples and (**B**) single individual samples from the four colonies. Symbols representing the four colonies represented in (**B**) can be found in maroon in (**A**). (C) Ranked Self-similarity S for each sample type comparison. Self-similarity is the absolute value of the difference between dissimilarity within and across samples divided by the average dissimilarity of all samples (by standardized Euclidean distance of protein abundance). Samples with higher S are more similar to samples of the same type, while samples with an S of zero are equidistant to the centroids of the two sample groups.

The online version of this article includes the following source data for figure 3:

**Source code 1.** Matlab source code to produce self-similarity scores, plots and PCA plots that make up **Figure 3**, https://github.com/dradri/variation2021.

**Source data 1.** Matlab MAT data file based on iBAQ values, gene names, and sample classes to produce self-similarity scores, plots and PCA plots that make up **Figure 3**.

sample volume (Pearson correlation test p < 0.03, $r = 0.24$ for colony samples and p < 0.01, $r = -0.40$ for single-individual samples).

Field-collected samples exhibited more variable proteomes than did lab-collected samples (**Figure 2b**, gamma GLM posthoc p-values < 0.001, **Figure 2—source data 1**). Further, colonies that had been in the lab for more than one year showed less variable proteomes than did colonies that had been in the lab for only six months (**Figure 2b**, gamma GLM $z = -4.46$, SE = 0.04, p < 0.001). The trophallactic proteome variability of young and mature colonies did not differ significantly, nor did nurses' and foragers' (**Figure 2b**). Foragers had fewer identified proteins in their trophallactic fluid than did nurses (**Figure 2a**, negative binomial $z = 3.72$, SE = 0.08, p = 0.005), and there were no significant differences among the main full colony samples (**Figure 2—source data 2**).

When the principal components of the trophallactic fluid proteomes were analyzed, the samples tended broadly to align with others of the same type, although clusters were not fully distinct (**Figure 3A**

*and B*). We developed a metric, self-similarity (S), to assess the depth of difference within and across sample types (*Figure 3C*). Because field-collected samples had more diverse protein content even within sample types (*Figures 2 and 3A*), the self-similarity in the *Young vs. Mature* comparison is low (*Figure 3C*). Single individual samples, and especially forager samples were less complex, allowing a larger proportion of their dissimilarity to be explained by sample type. Further, because our classification of nurse and forager is based on the individuals' location on brood or out-of-nest, it is possible that some nurse-classified individuals were either misclassified, transitioning from nurse to forager, or had trophallactic fluid in their crop uncharacteristic of their behavioral caste.

## Comparisons of trophallactic fluid across conditions

In addition to characterizing the most abundant and core proteins of the trophallactic fluid (*Figure 4*, *Figure 4—figure supplement 1*), we wanted to robustly identify proteins that differ significantly in our comparisons despite the noise inherently present in this social fluid. To accomplish this, we chose to overlay three distinct statistical approaches (*Figure 1B*): classical frequentist, empirical Bayes and machine-learning in the form of random forest classification. In our main comparisons, *Young vs. Mature* colonies from the field, young colonies in the *Field vs. Lab*, and individual *Nurses vs. Foragers* in the lab, we found significant differences between groups with all three analysis methods (*Figure 5*, *Figure 5—figure supplement 1*, *Figure 5—figure supplement 2*, full results for the significantly differing proteins in *Supplementary file 2* and for all proteins in *Supplementary files 3-5*).

For the *Young vs. Mature* comparisons, there were 10, 10, and 30 differentially abundant proteins according to frequentist t-test, empirical Bayesian LIMMA and the random forest approach, respectively. Similarly, for the *Nurse vs. Forager* comparison there were 21, 57, and 26 differentially abundant proteins, and when young colonies were brought to the laboratory and resampled after six months, 17, 31, or 29 proteins had significantly different abundance. The average accuracies of classification for comparisons with the random forest approach were: *Young vs. Mature*, 87 %; *Nurse vs. Forager*, 93 %; and *Field vs. Lab*, 91 %. This indicates that our trained classifier can predict whether a trophallactic fluid sample originates from a nurse or a forager with 93 % accuracy. We found no clear signature of spatial structure (*East vs. West*) in the trophallactic fluid proteomes. The frequentist analysis between different sampling areas found no significantly different proteins, and the random forest model did not reach high enough accuracy for this dataset to be informative (58 % classification accuracy). Only the empirical Bayes approach found eight proteins that significantly differed between the sampling areas (*Figure 5—figure supplement 1*, *Supplementary files 2 and 4*).

To leverage the unique benefits of the different forms of analysis, we focused our further analyses on proteins significantly different in two out of the three forms of analysis. Here, young and mature colonies differed by 12 proteins, and nurses and foragers differed by 19 proteins (*Figure 5*). When young colonies were brought to the laboratory and resampled six months later, the trophallactic fluid proteomes differed significantly by 20 proteins. Additionally, the single individual dataset showed that proteomes are affected both by colony of origin and by behavioral role of the individual, with 60 proteins showing significant interaction between the two factors (*Supplementary file 3*).

## Functions of the proteins in trophallactic fluid

To investigate the functions of the proteins found in trophallactic fluid, we performed functional enrichment analysis of gene ontology terms, pathways and protein-protein interaction (PPI) networks of the trophallactic fluid proteins' *Drosophila melanogaster* orthologs. The 60 most abundant proteins in trophallactic fluid (*Figure 4*, *Figure 4—figure supplement 1*) are predominantly involved in the biological processes of carbohydrate metabolism, lipid and sterol transport (*Figure 6*, *Figure 6— source data 1*, FDR < 0.00038, FDR < 0.0013 and FDR < 0.0087 respectively). The larval serum protein complex was represented by three out of four members in both the most abundant proteins and in the significantly differing proteins (hexamerins/arylphorins: Lsp1beta, Lsp1gamma, and Lsp2). A strong representation of the innate immune system (Reactome pathway FDR < 6.57e-5) was evident as were lysosomal processes (KEGG pathway, FDR < 3.21e-9).

Beyond the 60 most abundant proteins in trophallactic fluid, many others are of interest as well. A critical protein in insect physiology, vitellogenin is the 93[rd] most abundant protein in trophallactic fluid, present in 77% and 88% of colony and single individual samples, respectively. Three of the 60 most abundant proteins had no similarity to *Drosophila* genes, and thus could not be included in the

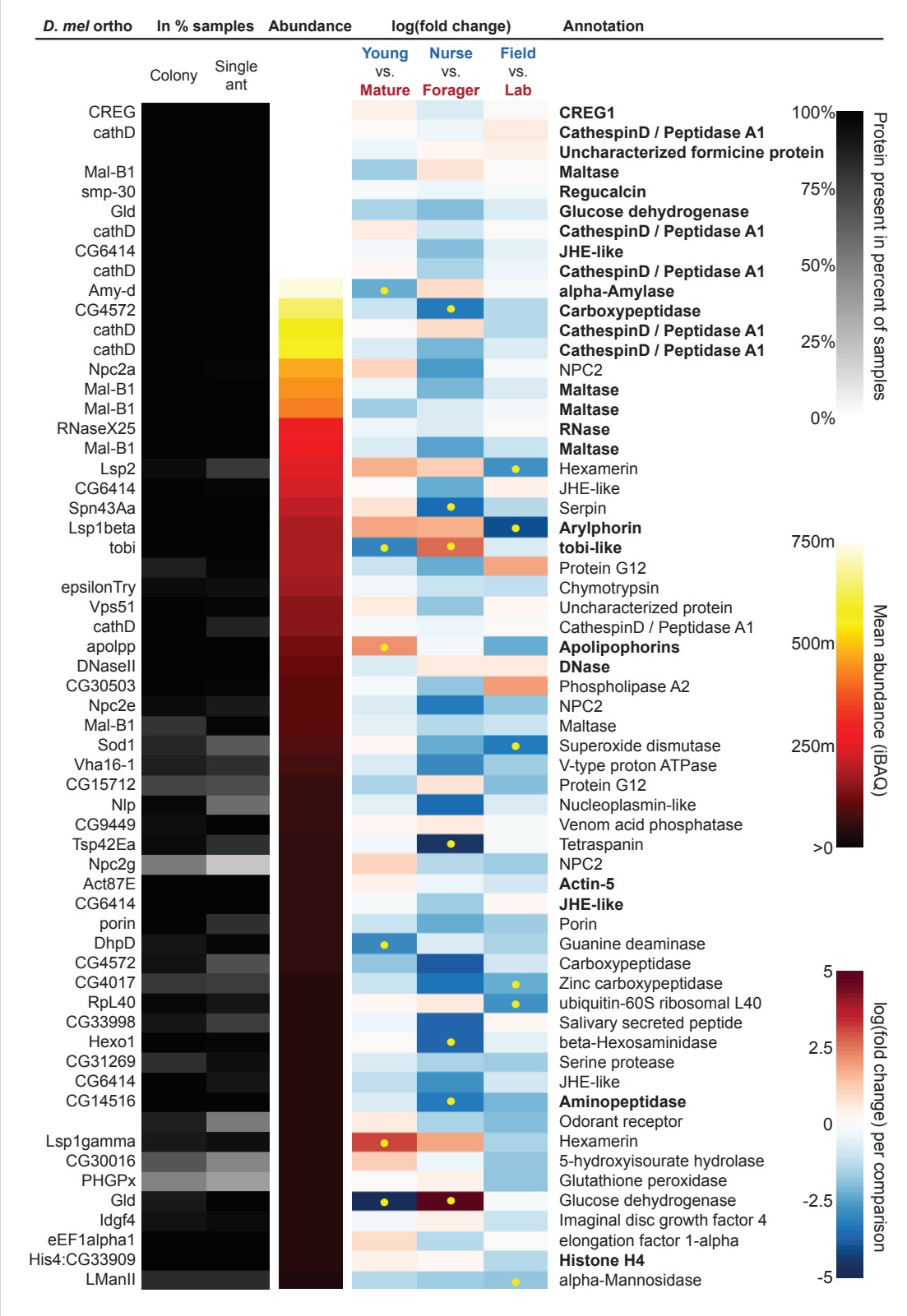

**Figure 4.** The sixty most abundant proteins in trophallactic fluid over 73 colony and 40 single individual samples. Ranking of abundance (including missing values). From left to right, *Drosophila melanogaster* orthologs, proportion of samples in which the protein was identified in colony samples and single individual samples, average iBAQ abundance across all samples, log2 of the fold change in abundance between types for a given comparison, the comparisons for which the protein was significant in two out of three methods are marked with yellow dots, annotation terms. Annotation terms

*Figure 4 continued on next page*

*Figure 4 continued*

are bolded for the 25 out of 27 core trophallactic fluid proteins that are amongst the 60 most abundant proteins. The additional but less abundant core proteins are a cathepsin (26–29 p) and a myosin heavy chain (Mhc). For protein accession numbers, see *Figure 4—figure supplement 1*.

The online version of this article includes the following figure supplement(s) for figure 4:

**Figure supplement 1.** Most abundant proteins with accession numbers.

functional enrichment analysis. One of them is a putative odorant receptor, another a G-protein alpha subunit, and the third showed no orthology to characterized proteins. None of these proteins significantly differed in more than one analysis for a given comparison.

Many of the trophallactic fluid proteins, abundant or significantly differing, were represented in trophallactic fluid by multiple genes from the same protein family, in some cases part of tandem repeats in the genome, indicative of relatively recent evolution. Multiple proteins of the same family were found in the most abundant trophallactic fluid proteins (*Figure 4*, *Figure 4—figure supplement 1*): a family of cathepsinD-like proteins (six in the top 60; *LeBoeuf et al., 2016*; *Hamilton et al., 2011*) and a family of Maltase-B1-like proteins (five in the top 60). In the list of significantly differing proteins (*Figure 5*, *Figure 5—figure supplement 2*), we observed fewer members of these families and instead saw three guanine deaminase proteins, all of which significantly differed in the *Young vs. Mature* comparison. Other families that showed duplications were glucose dehydrogenases, CREG1 and tobi-like proteins (target-of-brain-insulin).

There was an overlap of 16 proteins between the most abundant proteins and the proteins significant in two out of three of our statistical methods in any of the comparisons. The PPI network for our differentially abundant protein set (46 proteins, *Figure 5*) was similar to that of the most abundant proteins (*Figure 4*) but with increased interaction in the networks of the proteins themselves beyond what would be expected by chance (PPI enrichment p-value < 2.35e-11 in differentially abundant proteins relative to p-value < 1.59e-9 in abundant proteins), with noted enrichment in oxidation-reduction processes (FDR < 0.0026) and stronger enrichment in carbohydrate metabolic processes (FDR < 2.15e-6).

To better understand the functions of the significantly differing proteins in each comparison, we analyzed the GO terms and PPI networks of proteins significant in two out of three statistical methods separately for each of our three main comparisons (*Figure 6*, *Figure 6—source data 1*). The *Nurse vs. Forager* comparison yielded a network of proteins with more interaction than would have been predicted by chance (PPI enrichment p-value < 2.57e-4) as well as a higher degree of PPI enrichment than the other two comparisons (*Young vs. Mature* p < 0.002 and Field v Lab p < 0.02). The orthologs of differentially abundant proteins found in the behavioral caste comparison involved not only carbohydrate processing (FDR < 1.7e-4), but also oxidation-reduction and malate metabolic processes (FDR < 0.023 and FDR < 0.02, respectively). These pathways have been implicated in the determination of lifespan (*Wiley and Campisi, 2016*). Indeed, two of the 46 differentially abundant proteins over all comparisons have *D. melanogaster* orthologs with the gene ontology term 'determination of adult lifespan' (Men, Sod1). The *C. floridanus* tetraspanin, significantly more abundant in nurse trophallactic fluid, is a one-to-many ortholog to the family of Tsp42E genes, one of which has also been implicated in determination of adult lifespan in *D. melanogaster*.

As trophallactic fluid samples of young and mature colonies were distinguishable by principal component analysis and our random forest classifier, we wanted to see if our trained classifier could assess a change in maturity of our young colonies after they had spent six months in the laboratory. Our random forest classifier assigned an average out-of-box maturity score to our 16 laboratory samples of 42 % mature, reflecting the intermediate position of the laboratory colony samples in *Figure 3*.

## Discussion

When an ant colony matures, the protein composition of trophallactic fluid changes in biomarker-like manner, suggesting that these proteins circulating amongst individuals play a role in age-related colony metabolism and physiology. At the individual level, certain trophallactic fluid proteins correlate with behavioral caste within the colony, a trait known to encompass both individual task requirements and age (*Korb et al., 2021*; *Mersch et al., 2013*; *Wild et al., 2021*). Trophallactic fluid complexity

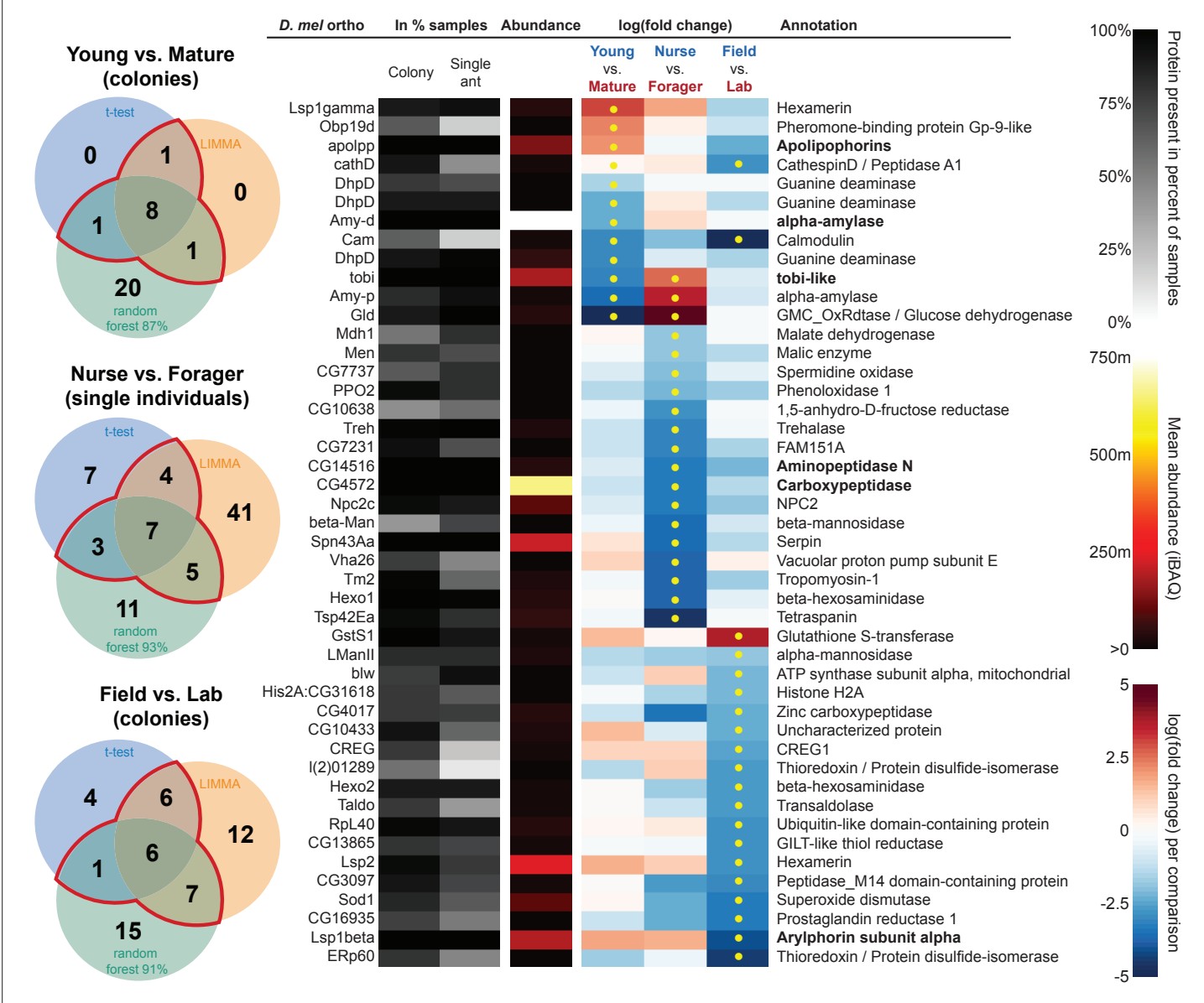

**Figure 5.** All proteins that significantly differ in two out of three of the analysis methods (frequentist, empirical Bayes, and random forest classification with SHAP values). From left to right, Venn diagrams of significance overlap between methods, *Drosophila melanogaster* orthologs, proportion of samples in which the protein was identified in colony samples and single individual samples, average iBAQ abundance across all samples calculated without missing values, log2 of the fold change in abundance between types for a given comparison, the comparisons for which the protein was significant in two out of three methods are marked with yellow dots, annotation terms. Annotation terms are in bold for the core trophallactic fluid proteins present in all samples. For visualization of each analysis method, see *Figure 5—figure supplement 1*. For protein accession numbers, see *Figure 5—figure supplement 2*. For all the 135 proteins significantly differing in any analysis, see *Supplementary file 2*. For full model results, see *Supplementary files 3-5*.

The online version of this article includes the following source code and figure supplement(s) for figure 5:

**Source code 1.** Jupyter notebook to run random forest analyses, https://github.com/dradri/variation2021.

**Figure supplement 1.** Visualization of all results.

**Figure supplement 2.** Significantly differing proteins in two out of three analyses with accession numbers.

declines over time when colonies are brought from the field to the laboratory. This may reflect dietary, microbiome or environmental complexity – typical of traits that have evolved to deal with environmental cues and stressors (e.g. immunity, *Lazzaro and Little, 2008*).

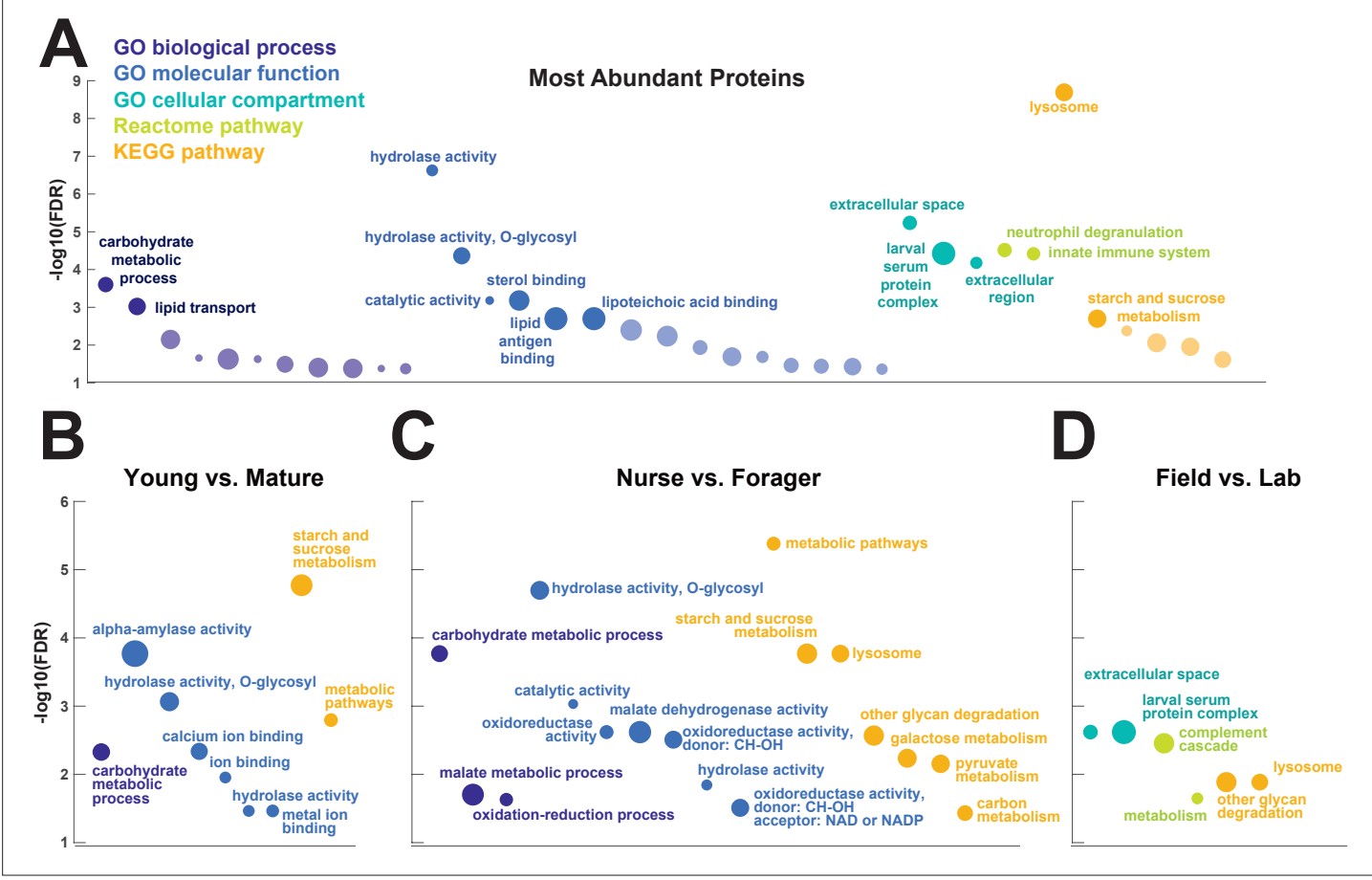

**Figure 6.** Gene set enrichment analysis of trophallactic fluid. Significant terms for *Drosophila melanogaster* orthologs of (**A**) the 60 most abundant trophallactic fluid proteins, trophallactic fluid proteins significantly differing between (**B**) *Young vs. Mature*, (**C**) *Nurse vs. Forager*, and (**D**) *Field vs. Lab*, with -log$_{10}$(FDR) indicated on y-axes. Deep purple indicates GO biological process; blue, GO molecular function; turquoise, GO cellular compartment; lime green, Reactome pathway; orange, KEGG pathway. Circle size indicates strength, log$_{10}$(observed proteins / expected proteins in a random network of this size). Full results can be found in *Figure 6—source data 1*.

The online version of this article includes the following source data for figure 6:

**Source data 1.** Gene set enrichment analyses of trophallactic fluid proteins.

Overall, our data reveal a rich network of trophallactic fluid proteins connected to the principal metabolic functions of ant colonies and their life cycle. Pinpointing contexts that induce changes in trophallactic fluid, along with the exact targets and functions of the proteins, are important subjects for future work. Our establishment of biomarkers transmitted over the social circulatory system that correlate with social life will allow researchers to formulate and test hypotheses on these proteins' functional roles.

## Metabolism changes with maturity

We found that trophallactic fluid includes many enzymes involved in metabolism and protein products of metabolism. Many are core trophallactic fluid proteins present in all samples, but many also differ significantly among the colony and individual life stages. Some proteins abundant in mature colonies (Lsps, apolpp *Burmester and Scheller, 1999*; *Burmester, 2002*; *Burmester, 1999*) are major insect nutrient storage proteins (*Burmester, 1999*) that may be required to consolidate resources into large workers and sexuals, potentially acting as superorganismal hormones. Proteins abundant in foragers and young colonies (Gld, tobi, Amy, Mal, *Buch et al., 2008*; *White et al., 2021*) are well-conserved enzymes for fast sugar processing. This suggests a functional role of trophallactic fluid in the social physiology of ant colonies.

Similar shifts in protein composition or gene expression can be seen in different tissues of multi-cellular organisms as life-stage priorities change, for example in the midgut of *drosophila* females after mating, where changes in expression are observed in many genes orthologous to the proteins we found here (*White et al., 2021*). Additionally, *Drosophila* larval hemolymph proteome changes as development unfolds (*Handke et al., 2013*), and many of these same proteins also appear in our comparisons of worker trophallactic fluid. We suggest that regulation of larval development may at least in part occur over the social network of ants, in line with previous experimental results (*LeBoeuf et al., 2018*).

## Ageing and division of metabolic labor

Viewing the colony as a superorganism, the division of reproductive labor between different types of workers (soma) and queens (germline) should result in different individuals requiring differing resources and sustaining differing metabolic costs. Our results support this hypothesis. We show that trophallactic fluid transmits numerous factors linked to ageing and coping with oxidative stress, including two of the three most well-known antioxidant enzymes: superoxide dismutase and gluta-thione peroxidase (*Monaghan et al., 2009*). These and other ageing-related proteins, such as those in redox pathways and malate metabolism (*Wiley and Campisi, 2016*; *Koch et al., 2021*), are especially elevated in nurses, the individuals that are physically the closest to the brood and queen in the trophallactic network.

These results link trophallactic fluid to one of the main topics of evolutionary ecology: the longevity-fecundity tradeoff between reproduction and coping with oxidative stress (*Monaghan et al., 2009*; *Flatt, 2011*; *Edward and Chapman, 2011*). Social insect individuals seemingly escape this tradeoff with long-lived and highly reproductive queens and short-lived, non-reproductive workers (*Monaghan et al., 2009*; *Edward and Chapman, 2011*; *Heinze and Schrempf, 2008*). We reveal a possible distributed metabolism which could explain why social insects seem to subvert this tradeoff. If molecules dealing with oxidative stress, or beneficial products of metabolism (nutrient storage proteins) can be spread over the circulatory system, as our results show, certain individuals may bear the costs that others in the network incur. This could account for some of the puzzling results on the plasticity of senescence in social insects (*Kramer et al., 2021*; *Heinze and Giehr, 2021*; *Lucas et al., 2019*), and provides a new perspective to analyze the regulatory changes of social insect reproductive castes with regard to ageing (*Korb et al., 2021*; *Negroni et al., 2019*; *Elsner et al., 2018*; *Corona et al., 2005*; *Gstöttl et al., 2020*; *Corona et al., 2016*; *von Wyschetzki et al., 2015*). While most previous work has focused almost exclusively on gene expression, we show that for species that engage in trophallaxis, expression studies are necessary but insufficient to understand where in the colony the relevant genes act.

Our gene-set enrichment analysis showed significant enrichment in immunity-related proteins characteristic of phagocytic hemocytes (*Shokal and Eleftherianos, 2017*) in trophallactic fluid ('innate immune system', 'complement cascade', 'neutrophil degranulation'). These results indicate that hemocytes may themselves be transmitted mouth-to-mouth, and generally shows the involvement of the social circulatory system in colony-level immune responses with implications for social immunity. Our results do not show clear caste differentiation in the abundance of immune-related proteins, as did a study in honey bees in glands that produce trophallactic fluid proteins (*Vannette et al., 2015*), though we do see similar regulation of sugar processing enzymes and glutathione-S-transferases.

## Evolution of trophallactic fluid

Trophallactic fluid is one of many social fluids in biology – milk and seminal fluid are similar examples of direct transfers of biological material between individuals. Such socially exchanged materials often contain molecules that target receivers' physiology beyond the fundamental reason for the transfer (*Bromfield et al., 2014*; *Savino et al., 2013*), and allow social effects to directly influence the evolutionary process as indirect genetic effects (*Linksvayer, 2015*; *McGlothlin et al., 2010*; *Wolf et al., 1998*). Some of the proteins we find to be significantly differing in our comparisons have previously been implicated in these other social transfers. For example, one of our protein hits is orthologous to *Drosophila*'s CG10433, a seminal fluid protein (*Findlay et al., 2008*) that impacts juvenile-hormone-associated hatch-rate post-mating (*Liu et al., 2014*). In another parallel to a phylogenetically distant social fluid, trophallactic fluid's most abundant protein is CREG1, a secreted growth-associated

glycoprotein also abundant in mammalian milk (*Zhang et al., 2015*). Finding molecular parallels in distinct behavioral processes hints at the fundamental role of these exchanges in the evolution of social physiology, and possibly common adaptive requirements for bioactive social fluids.

Lysosomal pathways are enriched in our most abundant trophallactic fluid proteins and in our set of significantly varying trophallactic fluid proteins between nurses and foragers, according to the KEGG analysis. Lysosomes are acidic and can be major players in secretion, autophagic flux and exocytosis (*Tancini et al., 2020*; *Csizmadia et al., 2018*; *Maruzs et al., 2019*) – processes that may be important for nurses that feed larvae by trophallaxis. These significant lysosomal signatures we see in trophallactic fluid may indicate the mechanism of secretion (*Martínez et al., 2020*), or may give us cues of how this fluid has evolved. As trophallactic fluid has become acidified in formicine ants (*Tragust et al., 2020*), lysosomal genes could have been duplicated and neofunctionalized to a new role in this acidic fluid, similarly to juvenile-hormone-esterase-like proteins in trophallactic fluid (*LeBoeuf et al., 2018*). The fact that many abundant trophallactic fluid proteins represent clusters of related proteins from a few families (cathepsins, guanine deaminases, maltases) suggests there has been adaptive evolution in the proteins arriving in this fluid.

## Conclusions

We show that the protein composition of ant trophallactic fluid varies across different external contexts and internal conditions both at the colony and at the individual level, suggesting that the dynamic trophallactic fluid proteome has key functions in social physiology and life cycle of colonies. By describing the natural variation of trophallactic fluid we have laid the groundwork for future studies on the possible functions of these proteins in controlling the colony life cycle, senescence, and behavior.

## Materials and methods

**Key resources table**

| Reagent type (species) or resource | Designation | Source or reference | Identifiers | Additional information |
|---|---|---|---|---|
| Other | UniProt Reference proteome (*Camponotus floridanus*); accessed February 2020 | UniProt | UP000000311 | |
| Other | NCBI RefSeq Reference proteome (*Camponotus floridanus*), v7.5 | NCBI RefSeq | GCF_003227725.1 | |
| Biological sample (*Camponotus floridanus*) | Trophallactic fluid (see details in *Supplementary file 1*) | This paper | *Supplementary file 1* | *Supplementary file 1* |
| Software, algorithm | MaxQuant v1.6.2.10 | MaxQuant | RRID:SCR_014485 | |
| Software, algorithm | Perseus v1.6.15.0 | Perseus | RRID:SCR_015753 | |
| Software, algorithm | R 3.6.1 | R | RRID:SCR_001905 | |
| Software, algorithm | Matlab 2020b | Mathworks | RRID:SCR_001622 | |
| Software, algorithm | R-package MASS 7.3–53 | R Project | RRID:SCR_019125 | |
| Software, algorithm | R-package LME4 | R Project | RRID:SCR_015654 | |
| Software, algorithm | R-package multcomp 1.4–15 | R Project | RRID:SCR_018255 | |
| Software, algorithm | LIMMA-pipeline-proteomics pipeline 3.0.0 | GitHub | 10.5281/zenodo.4050581 | |
| Software, algorithm | sklearn v0.22.1 | Scikit-learn | RRID:SCR_019053 | |
| Software, algorithm | Python 3.7.6 | Python | RRID:SCR_008394 | |
| Software, algorithm | SHapley Additive ExPlanations package v0.37.0 | GitHub | RRID:SCR_021362 | |
| Software, algorithm | OMA Browser | OMA Browser (*Martínez et al., 2020* release) | RRID:SCR_011978 | |

*Continued on next page*

*Continued*

| Reagent type (species) or resource | Designation | Source or reference | Identifiers | Additional information |
|---|---|---|---|---|
| Software, algorithm | Flybase | Flybase | RRID:SCR_006549 | |
| Software, algorithm | STRING v11 | STRING | RRID:SCR_005223 | |

## Study species

*Camponotus floridanus* is a common species of carpenter ant in the south-eastern USA, and has already been the focus of previous trophallactic fluid analyses (*LeBoeuf et al., 2016*; *LeBoeuf et al., 2018*). They live in dead wood or in man-made structures, often in urban habitats, and forage for honeydew, floral nectar, extra-floral nectar, and arthropod prey. Each colony has a single, singly mated queen (*Gadau et al., 1996*), and polydomous nest-structures where queenless satellite nests are common. Colonies grow to tens of thousands of workers and produce sexual brood only after multiple years of initial growth. Large established colonies have two morphologically differentiated worker castes, with variably sized small-headed minors focusing on brood care when young and foraging when old, and big-headed majors that engage in nest defense, foraging and food storage (*Deyrup, 2017*).

## Colony and sample identification

The species was identified based on worker and queen morphology (*Deyrup, 2017*; *Deyrup, 2003*; *Moreau et al., 2014*). In line with previous studies, we use the name *C. floridanus* with the knowledge that the taxonomy and nomenclature of the C. *atriceps* complex (to which it belongs) is not fully resolved (*Deyrup, 2017*).

We collected full young colonies (0–80 workers) and mature colony extracts (30–200 workers) on several Florida Keys islands (*Figure 1* and *Supplementary file 1*) in winter 2019 and 2020. A colony was deemed 'young' if the worker population was <100, primarily minors, and the queen was found (meaning both that the species could be clearly identified and that the nest was not a queenless satellite of an established colony), and 'mature' if the colony was larger ( > 1000 individuals visible) and the opened nest contained many large aggressive majors. Young colonies lack majors (*Gibson, 1989*) and individuals are generally less aggressive. We only collected mature colony samples when we also found larval brood in the opened nest. In our study area, we observed that young colonies are typically found nesting in different material than are mature colonies. Young colonies are often found under stones or in lumps of clay-like mud associated with crab burrows a short distance from the water, whereas the mature colonies were found nesting in large pieces of damp rotting wood.

## Laboratory rearing

Young colonies were brought to the lab and maintained in fluon-coated plastic boxes with a mesh-ventilated lid, at 25 °C with 60 % relative humidity and a 12 hr light/dark cycle. Each colony was provided with one or more glass tube for nesting, 10 % sugar water, and a Bhatkar & Whitcomb diet (*Bhatkar and Whitcomb, 1970*) and some *Drosophila melanogaster*. One week prior to proteomic sampling, we substituted the honey-based food with maple syrup-based food to avoid contamination with honeybee proteins (as in *LeBoeuf et al., 2016*).

## Trophallactic fluid collection

Field samples of trophallactic fluid were collected within eight hours of ant collection. Of the 20 young colonies and 23 mature colonies, workers collected from two of the mature colonies (L and N) were subdivided into six fragments to assess variation within a single colony (two samples from major workers, two samples from brood-associated workers, and two samples from the remaining minor workers). For all other analyses, only one of these for each colony (referred to as minors1) was used to avoid pseudo-replication. In the laboratory, the trophallactic fluid samples underlying the *Field vs. Lab* comparison were sampled after six months in the lab. The four colonies used for the single individual analyses had been in the lab for 18 months at the time of trophallactic fluid collection.

Trophallactic fluid was obtained from $CO_2$- or cold-anesthetized workers whose abdomens were gently squeezed to force them to regurgitate the contents of their crops. This method of collection

was shown previously to correspond to the fluid shared during the act of adult-adult stomodeal trophallaxis (*LeBoeuf et al., 2016*). For each colony, at least 30 individuals were sampled to obtain at least 10 µl of raw trophallactic fluid. For many young colonies only smaller samples were possible, because of the low number of workers (*Supplementary file 1*). Young colony samples were only used for further analysis if at least 2.5 µl of trophallactic fluid were collected. For single individual samples, workers with visibly full abdomens were chosen and the obtained sample volumes ranged from 0.7 µl to 2.2 µl. An individual was classified as forager, when it was seen outside the nest tube in the feeding area of an undisturbed laboratory nest box, and a nurse, when it remained in the nest tube even after the tube was removed from the original laboratory nest and placed into a new one. For colonies from which individual samples were collected, a pooled sample was also taken from individuals that remained after individual sampling. Samples were collected with glass capillaries into 5 µl of 1 x Sigmafast Protease Inhibitor Cocktail (Sigma-Aldrich) with 50 mM Tris pH nine in LoBind eppendorf tubes and were stored –80 C until further analysis. The total proteomics sample number is 73 colony samples of following types: 23 mature colonies with two of them sampled six times, 20 young colonies in the field, 16 young colonies in the laboratory, four laboratory colonies used for single individual sampling; and 40 individual samples: 20 nurses and 20 foragers.

## Protein mass spectrometry sample preparation and analysis

Samples were mixed with Laemmli sample buffer and pH was adjusted with 1 M Tris-Cl, pH 7. After reduction with 1 mM DTT for 10 min at 75 °C and alkylation using 5.5 mM iodoacetamide for 10 min at room temperature protein samples were separated on 4–12% gradient gels (ExpressPlus, GeneScript). Each gel lane was cut into small pieces, proteins were in-gel digested with trypsin (Promega) and the resulting peptide mixtures were processed on STAGE tips (*Rappsilber et al., 2007*; *Shevchenko et al., 2006*).

LC-MS/MS measurements were performed on a QExactive plus mass spectrometer (Thermo Scientific) coupled to an EasyLC 1000 nanoflow-HPLC. HPLC-column tips (fused silica) with 75 µm inner diameter were self-packed with Reprosil-Pur 120 C18-AQ, 1.9 µm (Dr. Maisch GmbH) to a length of 20 cm. A gradient of A (0.1 % formic acid in water) and B (0.1 % formic acid in 80 % acetonitrile in water) with increasing organic proportion was used for peptide separation (loading of sample with 0 % B; separation ramp: from 5 to 30% B within 85 min). The flow rate was 250 nl/min and for sample application 650 nl/min. The mass spectrometer was operated in the data-dependent mode and switched automatically between MS (max. of $1 \times 10^6$ ions) and MS/MS. Each MS scan was followed by a maximum of ten MS/MS scans using normalized collision energy of 25 % and a target value of 1,000. Parent ions with a charge state form z = 1 and unassigned charge states were excluded from fragmentation. The mass range for MS was m/z = 370–1750. The resolution for MS was set to 70,000 and for MS/MS to 17,500. MS parameters were as follows: spray voltage 2.3 kV; no sheath and auxiliary gas flow; ion-transfer tube temperature 250 °C.

The MS raw data files were uploaded into MaxQuant software (*Tyanova et al., 2016a*), version 1.6.2.10, for peak detection, generation of peak lists of mass error corrected peptides, and for database searches. MaxQuant was set up to search both the UniProt (RRID:SCR_002380, https://www.uniprot.org/) and NCBI (RRID:SCR_003496, https://www.ncbi.nlm.nih.gov/) databases restricted to *C. floridanus* (UniProt, February 2020 version; NCBI RefSeq, version 7.5), along with common contaminants, such as keratins and enzymes used for digestion. Carbamidomethylcysteine was set as fixed modification and protein amino-terminal acetylation and oxidation of methionine were set as variable modifications. Three missed cleavages were allowed, enzyme specificity was trypsin/P, and the MS/MS tolerance was set to 20 ppm. The average mass precision of identified peptides was in general less than one ppm after recalibration. Peptide lists were further used by MaxQuant to identify and relatively quantify proteins using the following parameters: peptide and protein false discovery rates, based on a forward-reverse database, were set to 0.01, minimum peptide length was set to 7, minimum number of peptides for identification and quantitation of proteins was set to one which must be unique. The 'match-between-run' option (0.7 min) was used, which helps improve the protein identifications especially for our single-individual samples. All proteins labelled as contaminants, reverse or only identified by site were excluded and proteins with scores less than 70 were removed. After the filtering, the dataset contained 519 proteins. Quantitative analysis was performed using iBAQ values.

Intensity-based absolute quantification (iBAQ) is the quotient of sum of all identified peptides and the number of theoretically observable peptides of a protein (*Schwanhäusser et al., 2011*).

## Statistical analyses

Analyses of dataset characteristics were performed in Perseus v1.6.15.0 (*Tyanova et al., 2016b*), R 3.6.1 (*R Development Core Team, 2013*) and Matlab R2020b (*Figures 2 and 3*). Differences in protein numbers among the sample types were analyzed with a negative binomial model, using the function nb.glm from the R-package MASS 7.3–53 (*Venables and Ripley, 2002*). Proteome variability per sample type, as measured by the coefficient of variation of the iBAQ abundance of each protein when present, was analysed with a generalized linear model with gamma distribution and log-link with the R-package LME4 (1.1–26) (*Bates et al., 2015*). The package multcomp 1.4–15 was used for post-hoc testing for both models. Pearson correlation tests were used to check whether obtained protein number correlates with the sample volume. Because significant correlation was found, all further analyses were done separately for the individual samples that have small volume, and colony samples that have larger volume. Principal component analysis was run in Matlab on raw iBAQ values, for both the individual and the colony datasets.

Metric for self-similarity (S) within and across samples was calculated in Matlab2020b (https://github.com/dradri/variation2021; *LeBoeuf, 2021*; copy archived at swh:1:rev:4a620922992272317f-3cedad3dae6e60871cb282) as follows: pairwise standardized Euclidean distances (dissimilarities, D) were calculated between each pair of samples based on square-root transformed and median subtracted protein abundances; these dissimilarities were averaged for each sample with the other samples within type $\bar{D}_{with\in}$ and with the samples of the other type $\bar{D}_{across}$ and divided by the average dissimilarity to all other samples. Thus, self-similarity was calculated as:

$$S = \left| \frac{\overline{D}_{within} - \overline{D}_{across}}{\overline{D}_{all}} \right|$$

To establish the proteins whose abundance differs significantly between sample types, samples were subdivided according to three main comparisons (*Figure 1*): *Young vs. Mature* colonies from the field, young colonies in the *Field vs. Lab* six months later, and individual *Nurses vs. Foragers* in the lab. In addition, the extent of spatial effects was analyzed for the field-collected *Young vs. Mature* dataset by dividing the sampling locations to two areas (*East vs. West*). For the colony data, the differing sample volumes may account for a small proportion of the significant differences in the *Young vs. Mature* comparison, and to lesser extent in the *Field vs. Lab* comparison, where sample volume is collinear with the sample type. Our analyses may miss some of the proteins more abundant in the young field collected colonies which have the smallest sample volumes.

Quantitative proteomic comparisons between sample types were performed independently with three different approaches to robustly identify significantly differing proteins: (*Szathmáry, 2015*) classical frequentist t-tests, (*Queller, 1992*) linear models with empirical Bayes variance correction, and (*Hamilton, 1963*) machine-learning paired with modified Shapley values. Our approach is designed to be at the same time conservative and to find most of the differing proteins among our comparisons of the trophallactic fluid. The frequentist t-tests are the most conservative, and they miss some interesting proteins due to their strict model expectations that allow only to use the most common proteins. The empirical Bayes approach to cope with sample variance is a more flexible method that allows use of the entire dataset, finding important hits also among the rarer proteins, although the high amount of missing values, where iBAQ equals zero, makes the model less powerful for these proteins (*Kammers et al., 2015*). The machine learning approach paired with modified Shapley values, although less well explored in the current proteomics literature, is promising for its ability to find multivariate patterns that the other methods miss, and results in interpretable classification. For each comparison, we report the full results of all three analyses in *Supplementary file 2* (significantly differing proteins only) and *Supplementary files 3-5* (all results). Our results and discussion sections focus on the proteins that appear significantly different based on two out of three analysis methods (*Figure 5*).

## Classical frequentist analysis

Within each dataset only proteins present in over 70 % of the samples were analyzed. Out of an original 519 proteins, the final datasets for each comparison contained the following number of proteins: *Young vs. Mature*, 172; *Field vs. Lab*, 137; and *Nurse vs. Forager*, 136. All data were log2 transformed and median-centered, and missing data were imputed by random sampling from normal distribution with 2SD downward shift and 0.3 width for each sample. For colony datasets, we used the permutation-based FDR of 0.05, and for the single individual dataset that contained more borderline-significant proteins, we used a more stable Benjamini-Hochberg FDR with a stricter threshold of 0.01. S0 parameter (similar to fold-change) was set to two for all analyses. All comparisons were run as two-sample t-tests, with the *Field vs. Lab* as paired.

For the individual dataset, the combined effects of colony identity and behavioral role (*Nurse vs. Forager*) and their interaction were analyzed with two-way ANOVA, with Benjamini-Hochberg FDR corrections performed in R with the base R 3.6.1 command 'p.adjust'. Both factors were also analyzed separately with multiple- and two-sample t-tests (for colony identity and behavioral role, respectively). To allow comparison to the other statistical methods, only the simple *Nurse vs. Forager* analysis without the interaction was used for combining the lists of significantly different protein abundances. Our balanced sampling guarantees the results of this simpler model are robust enough to find the most descriptive proteins for nurse and forager trophallactic fluid, even when the more complex interactive patterns are lost.

## Empirical Bayesian analysis

We implemented LIMMA (Linear Models for Microarray Data), a method for two-group comparison using empirical Bayes methods to moderate the standard errors across proteins (*Kammers et al., 2015*), on our score-filtered iBAQ proteomic datasets with the LIMMA-pipeline-proteomics pipeline 3.0.0 (http://doi.org/10.5281/zenodo.4050581) developed for R 4.0.2. Data were median-normalized before comparison and all comparisons were run with a log2 fold change cutoff of 2.

## Random forest and shap analysis

We used random forest models (sklearn.ensemble.RandomForestClassifier version 0.22.1 *Pedregosa et al., 2011*) to classify samples into one of two groups for each comparison. These analyses were performed in Python 3.7.6 in a Jupyter notebook (https://github.com/dradri/variation2021). For each comparison, 10 analyses were performed, each with a different seed. For each seed, the dataset was split into 80 % training set and 20 % test set, and a model was fit, tested and accuracy computed. If accuracy was below 85%, hyper-parameter tuning was performed with GridSearchCV (sklearn 0.22.1), and the model re-fit. A seed and its corresponding model were not retained for further analysis if accuracy could not be improved above 75 %. Accuracies for *East vs. West* ranged from 33 to 89% and over 20 seeds, only one could be improved above 75 %. The typical parameters: max_depth, 3 or 5; max_features, 'auto'; min_samples_leaf, 3; min_samples_split, 8 or 12; n_estimators, 100 or 500. Samples were classified with out-of-box scores (*Supplementary file 4*). The average accuracies of classification for comparisons were: *Young vs. Mature*, 87 %; *Nurse vs. Forager*, 93 %; *Field vs. Lab*, 91 %; *East vs. West*, 58 %.

To understand which proteins contributed to the classification, we used SHAP (SHapley Additive exPlanations, shap package v0.37.0 for Python 3), a game theory tool that explains the output of machine learning models (*Lundberg and Lee, 2017*). To analyze the importance of each protein in a given comparison (feature importance), we averaged the absolute value of the Shapley values per protein across the data to derive the feature importance. Then for each protein, we averaged the feature importances over each of the 10 seeded models. Proteins that have no impact on the model classification receive a feature importance value of 0. When ranked according to average feature importance, the data had an approximate Pareto distribution with an inflection point typically at feature importance of ~0.15. Thus, because there is no established cutoff for significance in this form of analysis, we chose to include as 'significant' in further analyses all proteins with a feature importance of >0.15 (*Supplementary file 5*).

For random forest predictions, models trained on the classification between young and mature colonies were used to classify the same young colonies after 6 months in the laboratory. Out-of-box scores were averaged over five seeded models.

## Orthology, gene ontology, and protein network analyses

Because little functional work has been done in ants, we analyzed gene ontology terms for the *Drosophila* orthologs to our genes of interest. Orthologs to *C. floridanus* trophallactic fluid proteins were determined with OMA ('Orthologous MAtrix' *Martínez et al., 2020* release *Altenhoff et al., 2021*). If no ortholog was found within OMA for a given gene, the protein sequence was protein BLASTed against *Drosophila melanogaster*. In some cases, no ortholog could be found. Annotations were compiled from NCBI RefSeq and UniProt annotations.

GO analysis was performed using both Flybase (*Larkin et al., 2021*) and STRING v11 (*Szklarczyk et al., 2019*). STRING was also used for protein-protein interaction and pathway analyses, including KEGG and Reactome (SI *Supplementary files 4 and 5*). The protein-protein interaction enrichment analysis in STRING used a hypergeometric test with Benjamini-Hochberg corrected FDR. Only 43 out of the 60 most abundant proteins had sufficient annotation for use by STRING while 44 of the 46 differentially abundant proteins had sufficient annotation.

## Acknowledgements

We thank Joanne Reiter for help with field work, Guillaume Kuhn for help with ant maintenance in the laboratory, and members of the Social Fluids lab and the Review Commons reviewers for their useful comments on the manuscript.

## Additional information

### Funding

| Funder | Grant reference number | Author |
| --- | --- | --- |
| Schweizerischer Nationalfonds zur Förderung der Wissenschaftlichen Forschung | PR00P3_179776 | Adria C LeBoeuf |
| Bundesbehörden der Schweizerischen Eidgenossenschaft | 2020.0228 | Sanja M Hakala |

The funders had no role in study design, data collection and interpretation, or the decision to submit the work for publication.

### Author contributions

Sanja M Hakala, Conceptualization, Data curation, Formal analysis, Funding acquisition, Investigation, Methodology, Resources, Validation, Visualization, Writing - original draft, Writing - review and editing; Marie-Pierre Meurville, Investigation, Methodology, Resources, Software, Visualization, Writing - review and editing; Michael Stumpe, Investigation, Methodology, Resources, Writing - review and editing; Adria C LeBoeuf, Conceptualization, Data curation, Formal analysis, Funding acquisition, Investigation, Methodology, Project administration, Resources, Software, Supervision, Validation, Visualization, Writing - original draft, Writing - review and editing

### Author ORCIDs

Sanja M Hakala  http://orcid.org/0000-0002-3762-623X
Marie-Pierre Meurville  http://orcid.org/0000-0001-6767-063X
Michael Stumpe  http://orcid.org/0000-0002-9443-9326
Adria C LeBoeuf  http://orcid.org/0000-0002-2931-1510

### Decision letter and Author response

Decision letter https://doi.org/10.7554/eLife.74005.sa1
Author response https://doi.org/10.7554/eLife.74005.sa2

# Additional files

## Supplementary files

• Supplementary file 1. The sampling scheme. Trophallactic fluid (TF) sampled for proteomics analysis. Date (field) indicates when the colony extract was collected from the field site, Date (TF sampling) indicates the date of the trophallactic fluid collection. Volume and ants indicate the volume collected and the number of ants collected from for each sample. The lab2019 colonies were used for single individual trophallactic fluid samples. For the two mature colonies that were sampled six times, * marks the sample that was used in the main datasets.

• Supplementary file 2. All 135 significantly differing proteins. This supplementary file combines into a single sheet the results and additional information for all of the significantly differing proteins in our four comparisons (Young vs. Mature, Nurse vs. Forager, Field vs. Lab, East vs. West), by all of the three statistical methods (classical, empirical Bayes, machine learning). Protein accession numbers, presence in colony and individual datasets, abundance when present, fold changes by comparison and significance both by comparison and by model are shared.

• Supplementary file 3. Full frequentist statistical results. Statistical results for the classical frequentist models; the imputed data are also shared.

• Supplementary file 4. Full empirical Bayes statistical results. For the empirical Bayes LIMMA models, results are shared as raw output tables.

• Supplementary file 5. Full random forest statistical results. Accuracy, seed, and mean feature importances for each gene are reported for each model trained for the random forest analyses.

• Transparent reporting form

## Data availability

The mass spectrometry proteomics data have been deposited to the ProteomeXchange Consortium via the PRIDE partner repository with the dataset identifier PXD028568. All other data are made available in this submission in figure supplements, source code and supplementary files.

The following dataset was generated:

| Author(s) | Year | Dataset title | Dataset URL | Database and Identifier |
|---|---|---|---|---|
| LeBoeuf AC | 2021 | Biomarkers in a socially exchanged fluid reflect colony maturity, behavior and distributed metabolism | https://www.ebi.ac.uk/pride/archive/projects/PXD028568 | PRIDE, PXD028568 |

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
