## [Decision Letter]

[Editors' note: this paper was reviewed by Review Commons.]

**Acceptance summary:**

This study significantly advances our understanding of how eusocial organisms communicate with each other within a colony by studying proteins in trophallactic fluids extracted from carpenter ants. These proteins help coordinate physiology and development among individuals in the hive. The methods used here are cutting edge and the work is expected to provide a foundation for many future studies.

---

## [Author Response]

Reviewer #1 (Evidence, reproducibility and clarity (Required)):The study was very well conducted by the group, selecting appropriated methods for achieving the aimed objectives. The sample were abundant and the statistical treatment were suitable for the size of samples, as well to compare different methods used in this study.The results in general were properly exploited by the authors, clearing many aspects of the role/function of the trophallaxis fluid. The results of this manuscript are apparently suggesting that young colonies prioritize the metabolization of carbohydrates, while mature colonies prioritize the accumulation and transmission of stored resources, amongst other processes. This study cleared many aspects about the role/function of the trophallaxis fluid for the colony.

We are happy the reviewer agrees with our choices of methods, sample sizes, and statistics, and we are pleased that they have come to the same conclusions.

Even considering the high level of present investigation, still there are some aspects that could be improved by the authors:– The text in general is relatively long with an over use of citations of literature;– The discussion is interesting, but sometimes too speculative; if the authorscould attenuate their speculative statements, the text would become more objective and fluid;

Thank you for this feedback. These comments truly helped us strengthen the manuscript. We have now streamlined the text, cutting down the introduction, cutting in half the discussion and we have made more explicit what is statement and what is speculation (more on this in response to reviewer 2).

– The results shown in figure 6A and 6D, relative to the processed of neutrophils degranulation and complement cascade, respectively. The authors did not discuss these results; is there a meaning at level of trophallaxis fluid role for the colony ? This was not discussed in the manuscript.

We thank reviewer #1 for pointing out these results. We have now addressed these terms in lines 277-284 of the discussion:

“Our analysis of gene-set enrichment analysis showed significant enrichment in immunity-related proteins characteristic of phagocytic hemocytes (58) in trophallactic fluid (innate immune system’, ‘complement cascade’, ‘neutrophil degranulation’). These results indicate that hemocytes may themselves be transmitted mouth-to-mouth, and generally shows the involvement of the social circulatory system in colony-level immune responses with implications for social immunity.”

– Considering the very high scientific quality of the present study, the authors could deposit all the raw proteomic data in a international reliable repository of proteins/DNA DB, since it will be required by top journals.

We wholeheartedly agree, and all data are now shared online through ProteomeXchange.

Reviewer #1 (Significance (Required)):Significance:The present investigation represents an important contribution for the knowledge the exchange of signals within the colony, to synchronize the physiology and development of the hive as whole the concept of superorganism.The existing data about the composition and potential role of the components from tropahallaxis fluid is very small, compared to the present results. The present study is a master piece of knowledge about the importance of eusociality.

Thank you for recognizing the importance of this study and affirming our work in such a wonderful way!

Audience:All those scientists involved with social insects; biochemists/protomists dedicated to insect biology, biochemistry and physiology.My expertise:Biochemistry of Arthropods secretion, in special of honeybees, ants and wasps.Referee Cross-commentingI think that both reviews aare complementary to each other; both reviews agree with the need to reorganize the text making it more compact and objective. Essentially, the authors must focus on the concept of trophallaxis. Thus, the biochemical processes outlined by proteomic analysis should be addressed to explain how the shared physiology of colony works out.

Our discussion now focuses more on trophallaxis as a whole, and the biomarker-like quality of the changing proteome. We agree the biochemical processes and their role in the shared colony physiology are fascinating topics. We have not yet performed follow-up experiments with the many proteins present in this fluid and thus do not want to over-conclude. We have now stated more clearly in the discussion what the current data can reveal about these topics, what is assumed via orthology, and what needs to be addressed in future studies.

Reviewer #2 (Evidence, reproducibility and clarity (Required)):This manuscript provides a comprehensive proteomic analysis of the trophallactic fluids extracted from carpenter ants. The analytical methods are state-of-the-art, and the results presented should fuel many studies. The vision of the research program, embodied in the title of the paper, is very exciting and is to be encouraged. However, the title of the paper in no way reflects the content of the paper, as none of the functional processes mentioned have been proven. This will require a lot of work and the development of perhaps new bioassays. I truly hope the PI’s lab takes this on a deep and substantial way; the notion of trophallaxis and its socially exchanged fluid has long captivated the fancy of social insect biologists, but with a few specific exceptions, the promise has not yet been realized. The technical and descriptive results presented here lay a strong foundation. For purposes of present publication, I strongly recommend a different title and a revised discussion that reflects the disconnect I outline. Cause/consequence issues need to be addressed.

We thank reviewer #2 for seeing our vision and that this is indeed foundational work that will “fuel many studies.” We also agree that the title and discussion contained too much speculation. The aim of this paper was to prove that there is systematic variation in trophallactic fluid in natural populations that correlates with biologically important social conditions, and further, that some proteins in this fluid can both act as biomarkers and be informative about underlying molecular processes. We have now communicated this more clearly in the introduction. In the revised version of the paper, we have reduced the speculation, and where appropriate, made it clear when there is speculation.

For example, discussion lines 233-238:

“Overall, our data reveal a rich network of trophallactic fluid proteins connected to the principal metabolic functions of ant colonies and their life cycle. […] Our establishment of biomarkers transmitted over the social circulatory system that correlate with social life will allow researchers to formulate and test hypotheses on these proteins’ <milestone-start /> <milestone-end /> functional roles.”

Three technical points:1) Sample sizes are low for some analyses (2/group) – though they are cleverly pooled.

We are not sure what the reviewer is referring to – none of our sample types had this low sample size (see SI Table 1 for sampling scheme). In contrast, for a proteomics study, our sample sizes are quite high. We are aware that for a study focusing on a natural population, the colony-level sample size of 16 (laboratory colonies) can be considered low, but this has been taken into account in our stringent statistical analyses.

2) How to distinguish between what animals actually transmit and what is found in the gut? There could be differences.

This has been addressed in our previous work, where it was shown that the crop content is equivalent to what is exchanged among individuals of this same species during the act of adult-adult stomodeal trophallaxis (Figure 1A, LeBoeuf et al. *eLife* 2016). We have now clarified this in the methods section of the current paper (line 361-364).

“Trophallactic fluid was obtained from CO_2_- or cold-anesthetized workers whose abdomens were gently squeezed to force them to regurgitate the contents of their crops. This method of collection was shown previously to correspond to the fluid shared during the act of adult-adult stomodeal trophallaxis (17).”

3) Is there evidence that the substances found are not just the product of digestion of ingested food? The differences between lab and field colony samples supports this.

In the type of proteomic analysis we have performed (the most commonly used proteomics approach when a genome is available), we detect only proteins found in the reference genome of interest (in our case *Camponotus floridanus*), so excepting cannibalism, we should not see proteins that originate from food. Note that this is why we do not provide lab colonies with the typical lab-reared ant diet that includes honey, as bees are also Hymenoptera, and royal jelly and trophallactic fluid have many proteins in common. Cannibalism could result in trace observation of many proteins, but could not produce the consistent and high-abundance set of proteins that we have observed as they are not produced in those precise ratios in larvae or adults.

The observed shift in trophallactic fluid from field to lab may reflect a change in diet or microbiome and these are questions that could be further investigated in future work (mentioned in lines 229-232). The clear difference we observe between trophallactic fluid of young and mature colonies, or the difference between the worker castes within a colony, is evidence that the variation observed in trophallactic fluid reflects more than diet.

“Trophallactic fluid complexity declines over time when colonies are brought from the field to the laboratory. This may reflect dietary, microbiome or environmental complexity – typical of traits that have evolved to deal with environmental cues and stressors (e.g. immunity, (37)).”

Reviewer #2 (Significance (Required)):The paper addresses a very important topic that should be of widespread interest to social biologists.Journal choice should reflect that this is a technically excellent paper that presents descriptive information but functional significance is highly speculative.

We appreciate that the reviewer agrees that our results are of widespread interest to social biologists. Indeed, our results must be somewhat descriptive, as we are working on a mostly unexplored socially exchanged fluid in a natural population. However, our study design tests clear hypotheses with preplanned sampling and experimental transfer of ant colonies to a new laboratory environment. We present confirmatory results of the hypothesis that trophallactic fluid is complex mixture of biomarker-like molecules and that these biomarkers can be used predict sample origin through machine learning (see random forest predictions, emphasized in lines 151-152). The fact that our evidence for this is correlative does not render it speculative. Indeed, in both ecology and in much of medicine, using correlative evidence is the norm, as it is often impossible to manipulate ecosystems, natural populations and some organisms in a safe and controlled manner. This is what convinced us to invoke the term ‘biomarkers’, <milestone-start /> <milestone-end /> as biomarkers are excellent examples of molecular correlates of larger conditions that have spurred advances in biology and medicine.

Some of the next steps in our research will be, as reviewer #2 suggested, additional studies on the roles of individual compounds of trophallactic fluid, building on the results of this paper. Additionally, while this study may not have explored the roles of specific molecules, open ended exploration is extremely important and necessary for any scientific advancement in the long run (*eLife* 2020;9:e52157).

All in all, we are grateful for this comment, as it showed us that we must communicate the aims of our work more clearly – which we have now done both in introduction (line 77-91) and throughout the discussion.

Referee Cross-commentingYes. Most of the discussion is pure speculation because we do t k ow what is exchanged and what the modes of action might be. But it's a great start!

We have reduced the speculation on the roles of single molecules, and we hope our responses to the points above clarify some of the reviewer’s uncertainties about what is exchanged. However, we do still outline hypotheses for potential functions and origins in the Discussion section, as this study is intended to be a foundation for new lines of research.